# Serum Sodium Concentration During Arginine Vasopressin Infusion in Critically Ill Children

**DOI:** 10.3390/children11111359

**Published:** 2024-11-09

**Authors:** Rafael Muff, Verena Gotta, Vera Jaeggi, Luregn J. Schlapbach, Philipp Baumann

**Affiliations:** 1Department of Intensive Care and Neonatology, University Children’s Hospital Zurich, University of Zurich, 8008 Zurich, Switzerland; rafael.muff@uzh.ch (R.M.); luregn.schlapbach@kispi.uzh.ch (L.J.S.); 2Children’s Research Centre, University Children’s Hospital Zurich, University of Zurich, 8008 Zurich, Switzerland; 3Department of Paediatric Pharmacology and Pharmacometrics, University of Basel Children’s Hospital, 4056 Basel, Switzerland; verena.gotta@ukbb.ch; 4Department of Paediatric Clinical Pharmacy, University of Basel Children’s Hospital, 4056 Basel, Switzerland; 5Department of Data Intelligence, University Children’s Hospital Zurich, University of Zurich, 8032 Zurich, Switzerland; vera.jaeggi@kispi.uzh.ch

**Keywords:** neonate, paediatric intensive care, sodium, antidiuretic hormone, vasopressin, arginine vasopressin, vasopressor, inotrope, hyponatraemia

## Abstract

Background: Intravenous arginine vasopressin is increasingly used for the treatment of critically ill children. It bears the risk of hyponatraemia with potential severe long-term sequelae, but data on hyponatraemia as a side effect of continuous vasopressin infusion for paediatric intensive care patients is scarce. Methods: In this retrospective analysis performed at a tertiary care paediatric intensive care unit with 2000 annual admissions, patients were included if they were treated with intravenous vasopressin between 2016 and 2022. Baseline sodium concentrations, lowest sodium concentrations during arginine vasopressin treatment, and time to lowest sodium concentration (nadir) were derived. Results: In total, 170 patients with a median age of 4 months [interquartile range, IQR, 0–33] were included, 92.4% underwent surgery, and 28.8% died. Median arginine vasopressin dose rate was 0.027 IU/kg/h [0.019–0.036] and arginine vasopressin was started 3.2 [0–26] h after intensive care admission. Median arginine vasopressin application duration was 13.6 h [6.2–32.6]. Baseline sodium was 141 mmol/L [138–145], and lowest median sodium during arginine vasopressin infusion was 137 mmol/L [132–141] (nadir at 8.4 h [1.0–28.1] after arginine vasopressin start). Hyponatraemia (<135 mmol/L) occurred in 38.2% of patients during AVP treatment, and physicians administered a median of 10.2 mmol/kg/d [6.2–16.4] sodium during arginine vasopressin therapy. Conclusions: Under arginine vasopressin infusion, hyponatraemia was common, although high daily doses of sodium were administered to keep the serum values in physiologic ranges. This emphasises the need for close electrolyte monitoring and sodium substitution in children and adolescents under arginine vasopressin treatment to avoid hyponatraemia and related sequelae.

## 1. Introduction

Critically ill paediatric patients are at high risk for arterial hypotension and shock during intensive care treatment [1]. According to current guidelines, arterial hypotension and shock should be treated early with intravenous fluids and vasopressors [2]. In this way, organ perfusion and thus oxygen and nutrient transport to the organs can be restored [3]. Although life-saving, fluid administration and catecholamines are associated with severe side effects, such as volume overload, electrolyte shifts, cardiac arrhythmias, increased cardiac afterload, and decreased peripheral perfusion [4]. Therefore, in the recent past, alternatives to volume and catecholamine therapy have been sought. Intravenous arginine vasopressin (AVP) infusion has emerged as one such potential alternative for the treatment of arterial hypotension apart from catecholamines. It is increasingly used in the routine treatment of critically ill neonates and children [5,6,7].

Arginine vasopressin is an endogenous peptide hormone with its main functions in body physiology being V1 AVP receptor-mediated peripheral vasoconstriction and V2 AVP receptor-mediated water reabsorption in the renal collecting duct [8]. Further, V3 AVP receptors in the anterior pituitary and in the pancreas steer the secretion of adrenocorticotropic hormone and insulin, respectively [9].

Hyponatraemia is an important potential side effect of arginine vasopressin treatment that results from increased water reabsorption compared to sodium reabsorption [10]. Hyponatraemia can have various serious consequences that might increase morbidity and mortality: lethargy, headache, and nausea may be the first symptoms of a severe encephalopathy, which can end in epileptic seizures, coma, and finally death [11]. Further, if hyponatraemia is corrected too quickly, cerebral dehydration might occur as water shifts from intracellular to extracellular spaces following osmotic gradients. This causes myelin degeneration, especially in the pons, with devastating consequences like progressive tetraspasticity and even death [12].

Hyponatraemia as a side effect of intravenous AVP therapy is poorly studied in the literature for critically ill children. The published courses of sodium values [10] did not reflect our clinical experience in the paediatric intensive care setting. As intensive care physicians, we regularly observed relevant decreases in serum sodium concentrations and increases in demand for exogenous sodium administration during AVP therapy. In the published literature, the necessary sodium substitutions were underreported.

This retrospective study aimed to analyse serum sodium concentration progression during arginine vasopressin therapy in critically ill paediatric patients, taking into account the amount of sodium supplied.

## 2. Methods

### 2.1. Settings and Participants

This retrospective study used data extracted from the electronic healthcare records of the mixed medical and surgical paediatric intensive care unit (PICU) of the University Children’s Hospital Zurich with 2000 annual admissions (Metavision^®^ (iMDsoft, Tel Aviv, Israel), Phoenix^®^ (Aveniq AG, Oftringen, Switzerland)). Metavision was the electronic monitoring and prescription chart in the PICU of the University Children’s Hospital Zurich and was used firstly to identify all eligible patients. The inclusion criteria were 1. hospitalisation at the PICU of the University Children’s Hospital Zurich in the years 2016–2022; 2. age between first day of life and 18^th^ birthday; 3. treatment with intravenous arginine vasopressin for any reason; and 4. general consent of the University Children’s Hospital Zurich signed by legal guardians of the children for further use of routinely stored health related data for research purpose in coded form. The further use of data for this project was approved by the local ethics committee (KEK 2021-02276). The exclusion criteria were 1. one of the above-mentioned inclusion criteria not fulfilled or 2. AVP treatment initiated beyond 21 days of PICU admission to focus on acutely ill children.

### 2.2. Outcome Measures

The primary outcome was the description of the course of the serum sodium concentration during AVP therapy, numerically summarized as follows: (1) Baseline: sodium 24 h prior to start of AVP therapy and % of patients with hypontraemia at baseline; (2) during AVP: % of patients with hyponatraemia (serum sodium <135 mmol/L), lowest serum sodium (nadir), and time to nadir; and (3) post-AVP: serum sodium concentration approximately 24 h (defined as 20–28 h) after cessation of AVP therapy. For patients with hyponatraemia during AVP treatment, the percentage of patients (%) with sodium correction was calculated along with time to reach physiologic levels again (during AVP treatment if observed or following cessation of AVP treatment).

The secondary endpoints were the course of sodium supplementation and fluid uptake during AVP therapy, the relationship of lowest serum sodium (nadir) with AVP mean individual dose, AVP cumulative dose, AVP treatment duration, and clinical parameters in critically ill neonates and children admitted to a tertiary care intensive care unit.

The following variables were extracted from Metavision and Phoenix systems for all individual patients and entered manually into RedCap^®^ (https://projectredcap.org, accessed on 14 October 2024) case report forms. Baseline parameters: age (months), weight (kg), height (cm), primary diagnosis leading to PICU admission (classified in six categories as cardiac, visceral, respiratory, shock, trauma, and other), surgery (if any, yes/no), Pediatric Index of Mortality II (PIM II, index range: 0–100), duration of invasive ventilation (hours), length of ICU stay (LOS, days), and death. For primary endpoint variables: AVP dose and treatment duration, and serum sodium 24 h before until 24 h (range 20–28 h) after AVP treatment. For secondary endpoint variables (24 h before, during, and 24 h after AVP therapy): total supplemental sodium uptake (amount prescribed in addition to baseline enteral nutrition or parenteral nutrition formulas). Fluids containing sodium included NaCl 0.9%, NaCl 3%, NaCl 25%, sodium bicarbonate, Ringer’s acetate, Ringer’s lactate, Ringer’s acetate-glucose 1%, glucose 4.6%/sodium 0.9%, glucose 9.1%/sodium 0.9%, and mixed 4:1 solution (sodium 31 mmol/L, glucose 80 g/L). Further, the total amount of daily intravenous fluids; peripheral oxygen saturation; transcutaneous CO_2_; heart rate; systolic, mean, and diastolic arterial blood pressure; near-infrared spectroscopy (NIRS); body temperature; potassium, chloride, calcium, and lactate concentrations; blood gas analysis values; and catecholamine, steroid, and diuretic doses.

### 2.3. Statistical Analysis

For the analysis, the coded data from the RedCap database were transferred to the IBM SPSS^®^ (IBM, Armonk, New York, NY, USA, version number 29.0.0.0 (241)) and R (R Version 4.2.1) statistics programmes. For each patient. the duration of AVP administration and cumulative AVP dose was calculated, and the mean individual AVP dose rate derived (cumulative AVP dose/duration of AVP administration). Sodium supplementation and fluid intake were calculated for each calendar day. For all other time-varying clinical variables, the mean individual recorded value per AVP dosing rate was calculated at baseline (24 h prior to AVP start), during AVP treatment, and 20–28 h after cessation of AVP treatment, to create descriptive summary statistics. Hyponatraemia was classified as mild/moderate/severe according to Spasovski et al. as 130–135/125–130/<125 mmol/L, respectively [13].

Baseline characteristics, as well as outcome variables before, during, and after AVP treatment were summarised as median [interquartile range] or number (%); missing data before or after AVP treatment were not imputed. For descriptive comparisons before, during, and after AVP treatment, the Kruskal–Wallis test was used and post hoc pairwise comparisons were performed using the Wilcoxon test. In a sensitivity analysis, the primary outcome variables were recalculated for a subset of patients with complete sodium information also before and after AVP treatment.

The dose–response relationship between the nadir of serum sodium and AVP treatment (mean individual AVP dose rate, cumulative AVP dose, and duration of AVP administration) was assessed in a linear regression analysis (considering variable transformations as appropriate following visual inspection). The following clinical potentially influencing factors on the sodium nadir were further considered in univariable and multivariable regression analyses: main clinical diagnosis (cardiac, shock, visceral, respiratory, trauma, or other), age, weight, PIM (<10 versus ≥ 10), mean individual hydrocortisone dose during AVP treatment, and mean individual furosemide dose during AVP treatment. Statistical significance level: *p* < 0.05.

Reporting was performed according to the STROBE checklist for reporting of retrospective cohort studies (STROBE-checklist-v4-cohort).

## 3. Results

### 3.1. Patients and AVP Administration

Two hundred and forty-four patients were identified as fulfilling the above-mentioned inclusion criteria: hospitalization at the ICU of the University Children’s Hospital Zurich in the years 2016 until 2022 between the first day of life and the 18^th^ birthday with administration of intravenous AVP. Among those patients, 181 patients had the general informed consent of the University Children’s Hospital Zurich for research with coded data signed by legal guardians. Eleven patients received AVP later than 21 days into their hospitalization at the ICU. Thus, 170 patients were included in this study. The baseline characteristics of the study population are presented in Table 1. One hundred and fifty-seven patients (92.4%) underwent any kind of surgery and forty-nine patients (28.8%) died. Median length of intensive care stay (LOS) was 21 days [8.8–48.2]. AVP was given for a median duration of 13.6 h [6.2–32.6], with a median AVP dose rate of 0.027 IU/kg/h [0.019–0.036] (median cumulative dose: 0.37 IU/kg [0.15–0.95]). For 95 patients, serum sodium 24 h prior to AVP start was available, and for 127 patients, data for 20–28 h following AVP cessation, respectively (resulting in a total of 59 patients with information on serum sodium concentration before, during, and after AVP treatment).

### 3.2. Primary Endpoint

Course of Serum Sodium Under AVP Infusion

At AVP treatment start, the serum sodium concentration for 170 study patients was 141 mmol/L [138–145] (with hyponatraemia already present in 21/170 (12.4%) of patients, and among them, with moderate degree in 4/170 (2.4%)). Median serum sodium concentration during AVP therapy was 139.1 mmol/l [135.1–143.0]. During AVP treatment, the lowest sodium concentration (nadir: 137 mmol/L [132–141]) was observed at 8.4 h [1.0–28.1] after the start of AVP treatment and 65/170 (38.2%) of patients experienced hyponatremia during AVP treatment (among them, moderate hyponatraemia in 20/170 (12.3%) of patients and severe hyponatraemia (<125 mmol/L) in 8 (4.7%) of patients). A summary of the course of serum sodium levels during AVP treatment is provided in Figure 1.

The incidence of hyponatraemia differed significantly before, during, and after AVP treatment (*p* = 0.007), and serum sodium concentrations < 135 mmol/L were observed in 29/95 (30.5%) of patients 24 h prior to AVP start (among them, moderate hyponatraemia (125 to <130 mmol/L) in 4/95 (4.2%) of patients) and in 24/116 (20.7%) 20–28 h after AVP treatment stop (moderate in 4 (3.4%) of patients). Correspondingly, also lowest sodium values differed significantly before, during, and after AVP treatment (*p* = 0.012), sodium concentrations 24 h prior to AVP start were 137 [134–140] mmol/L, and those 20–28 h after AVP treatment stop were 138 mmol/L [136–142]. A summary from the sensitivity analysis for the 59 patients with information on serum sodium for all time points (before, during, and after AVP treatment) is provided in Appendix A and shows overall similar results, albeit with even increased proportion of hyponatremia during AVP therapy in 27/59 (45.8%) of patients.

In 17/65 (26.1%) of patients with hyponatraemia during AVP infusion, correction of the serum sodium levels towards physiologic concentrations was observed during AVP treatment (time to correction 39 [6–95] h). In 25/40 (62%) of patients with hyponatraemia at the end of AVP treatment, correction of serum sodium to physiologic levels was observed within 20–28 h after AVP treatment stop.

### 3.3. Secondary Endpoints

#### 3.3.1. Sodium Administration and Fluid Uptake Under AVP Infusion

Hyponatraemia prevalence increased, although higher daily doses of sodium intake were given during arginine vasopressin infusion (Figure 2a). In total, physicians administered 10.2 mmol/kg/d [6.2–16.4] of sodium during AVP treatment. The daily fluid uptake during AVP infusion was recorded with a median of 153.3 mL/kg/d [95.8, 288.9]. Statistically significant higher daily doses of sodium and fluid were administered during AVP therapy than 24 h before or after (Figure 2). The net fluid balance, defined as the difference between fluid uptake and output, showed a median of +180 mL/d [0, 531].

#### 3.3.2. Association of AVP Dose and Treatment Duration with Lowest Serum Sodium Concentration

Figure 3 shows the association of lowest serum sodium measured during AVP infusion (nadir) with mean AVP rate, cumulative AVP dose, and duration of AVP infusion. There was no association with mean AVP rate (*p* = 0.41). Both cumulative AVP dose and duration of AVP treatment were associated with the nadir in the monovariable analysis (*p* < 0.001), while in the multivariable analysis, only the association with duration of AVP treatment remained significant (*p* = 0.03) (see Section 3.3.3). The estimated baseline sodium ≤12 h was 139 mmol/L (95%CI: 137.7–140.4). The estimated effect for duration >12 h (monovariable model) was a sodium decrease by −2.7 mmol/L (95%CI: −1.9 to −3.6 mmol/L) for doubling the duration of administration (e.g., to 24 h, 48 h, 96 h, and 168 h of treatment).

#### 3.3.3. Association of Other Clinical Parameters with Lowest Serum Sodium Concentration

Apart from AVP treatment duration, only low patient weight < 10 kg (decrease by −4.1 mmol/L for 50% reduction in weight < 10 kg, i.e., to 5 kg, and to 2.5 kg, *p* < 0.001) and the mean dose of the diuretic drug furosemide (decrease by −1.8 mmol/L for each 1 mg/kg of furosemide estimated, *p* = 0.007) were associated with the lowest serum sodium concentrations in the multivariable regression analysis (Figure 4). For the other covariates (main diagnosis, PIM </≥10, and hydrocortisone dose), no significant associations were found (*p* > 0.05,), and age < 12 months was highly correlated with weight < 10 kg and was therefore not included.

## 4. Discussion

This retrospective study aimed to analyse serum sodium concentrations during intravenous arginine vasopressin therapy in critically ill paediatric patients, taking into account the amount of supplemented sodium. Hyponatraemia under AVP infusion was indeed common (observed in 38% of patients, with moderate and severe hyponatraemia in 12.3% and 4.7% of patients, respectively) although high daily doses of sodium intake were administered to keep serum values in the physiologic range (median: 10.2 mmol/kg/d [6.2–16.4]). Moreover, we found decreased serum sodium levels to be associated with AVP treatment duration > 12 h, low patient weight, and furosemide administration in a dose-dependent manner.

### 4.1. Intravenous AVP Therapy and Hyponatraemia

Continuous intravenous AVP infusion has emerged for managing arterial hypotension in the routine care of critically ill neonates and children [5,6,7]. Further, AVP has been advocated by the Surviving Sepsis Campaign Guidelines for the management of septic shock as a second-line treatment, if catecholamines do not achieve sufficient perfusion [2]. However, reduced intravascular sodium levels are recognized as a significant potential complication of arginine vasopressin therapy due its effect on water reabsorption [10]. Acute hyponatraemia occurs when water reabsorption outpaces sodium reabsorption in the kidneys. Hyponatraemia can lead to various severe consequences for the human body, ultimately impacting mortality [11].

Despite its importance, hyponatraemia as a side effect of AVP therapy in critically ill children remains poorly studied in the literature. Davalos et al. examined the changes in serum sodium levels within 72 h of initiating intravenous AVP therapy in a retrospective analysis [10]. Patients receiving intravenous AVP exhibited significantly lower mean sodium levels within the first 72 h post-surgery than patients without AVP therapy (134.7 mmol/L (±3.8) vs. 137.1 mmol/L (±4.3), respectively). Further, they found notably increased incidences of hyponatraemia (48% vs. 17%, respectively), which is in line with our findings. The lowest median serum sodium levels during AVP infusion in our study were slightly higher with 137 mmol/L, however with a large range (lowest value ≈ 110 mmol/L, 4.7% of patients had severe hyponatraemia).

Alakeel et al. also investigated the incidence of hyponatraemia associated with AVP use in critically ill children, including a relevant cohort (65%) undergoing cardiopulmonary bypass. They reported significant decreases in mean sodium levels following AVP initiation as well with a serum sodium nadir of 134.3 mmol/L (vs. a nadir of 137 mmol/L in our study) [14]. This is consistent with recent case reports, which also report relevant decreases in serum sodium during intravenous arginine vasopressin therapy [15,16]. After AVP discontinuation, sodium levels slightly increased in both our study and the study by Alakeel et al. after the cessation of AVP. Hyponatraemia, defined as a serum sodium concentration below 135 mmol/L, was found in this study in 38.2% during AVP treatment, while the study by Alakeel et al. showed an increase in hyponatraemia incidence to 49.4% at 48 h into AVP administration. Further, both studies observed persistent hyponatraemia after the cessation of AVP. Both studies involved complex medication regimens, including diuretics and other vasoactive agents, which could contribute to variations in sodium balance and hyponatraemia risk. Loop diuretics inhibit the Na-K-2Cl carrier in the luminal membrane of the thick ascending limb in the loop of Henle and consecutively reduce the uptake of sodium and chloride into the cell. This results in increased natriuresis and diuresis [17]. As half of the patients in this study cohort received furosemide in parallel to AVP, this might have aggravated hyponatraemia. Accordingly, we found a significant association of furosemide with hyponatraemia in the multivariable regression analysis. On the contrary, 60% of the study cohort received steroids, mainly hydrocortisone, which was used at our PICU as second-line medication for catecholamin-refractory hypotension and shock. Mineralocorticoids lead to water and electrolyte retention [18] and might have diminished, at least partly, the hyponatraemia expression. It might well be that AVP use without steroids would result in even pronounced hyponatraemia states. The above-mentioned studies [10,14] did not provide detailed information on daily sodium supplementation to counteract sodium decreases and avoid hyponatraemia. In our study, significant amounts of additional exogenous sodium (10 mmol/kg daily) were administered to this study population to keep the serum sodium concentrations in the physiologic range.

Analysing fluid intake and fluid balance is just as significant for understanding the dynamics of hyponatraemia in paediatric intensive care units. In this study, the daily fluid intake during AVP therapy was recorded with a median of 153.3 mL/kg/d [95.8, 288.9]. Additionally, the net fluid balance, defined as the difference between fluid uptake and output, showed a median of +180 mL/d [0, 531]. Various mechanism might play a role leading to fluid overload during critical care treatment, such as post-surgical or septic systemic inflammatory response syndrome, syndrome of inappropriate antidiuretic hormone secretion, and iatrogenic excess of fluid administration for haemodynamic stabilization. Therefore, careful monitoring and cautious adjustment of fluid administration is essential to effectively minimize the risk of hyponatraemia and ensure optimal treatment outcomes. This is especially important in young children (in our cohort < 10 kg) who frequently react with very severe post-surgical inflammation, low cardiac output syndrome following cardio-pulmonary bypass surgery, and haemodynamic instability.

### 4.2. Limitations

While this study offers valuable insights into arginine vasopressin-associated hyponatraemia, the following limitations should be acknowledged. First, the retrospective character limited the ability to establish causal relationships or infer temporality between variables. Second, as a single-centre study without a control group, the findings may not be fully generalizable to broader populations due to variations in patient demographics, clinical practices, and resource availability across different healthcare settings. Third, the high rate of surgical interventions among this study population introduces potential bias related to the severity and complexity of underlying conditions, impacting the generalizability of the findings to all paediatric intensive care unit patients. Fourth, we could only include pre-AVP treatment sodium levels for 95 patients, as many patients were admitted to the PICU post-surgery and baseline sodium data availability for 24 h before AVP treatment start were limited. Last, the observed high mortality rate among patients receiving AVP therapy reflects the severity of illness in the study cohort, which may confound the association between AVP therapy and serum sodium levels due to underlying disease severity and associated complications such as systemic inflammatory response syndrome.

## 5. Conclusions

This retrospective analysis highlights the risk of hyponatraemia in critically ill paediatric patients receiving intravenous AVP for arterial hypotension. Despite the administration of high daily doses of sodium to maintain physiologic serum levels, more than half of the patients experienced significant decreases in serum sodium concentrations. These findings emphasize a critical need for careful electrolyte monitoring and management during AVP therapy to prevent hyponatraemia and its potentially severe neurological consequences. Future research with case–control studies would enhance the knowledge on hyponatraemia caused by intravenous AVP administration.

## Figures and Tables

**Figure 1 children-11-01359-f001:**
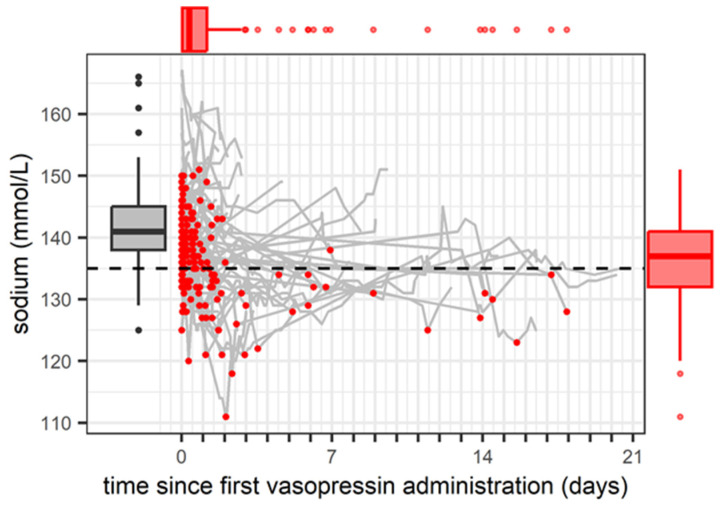
Course of serum sodium levels during arginine vasopressin infusion in critically ill children. Solid grey lines show the serum sodium concentration with individual trajectories over time; red dots are the lowest serum sodium measurements for each individual. Box plots show the interquartile range (IQR): Grey box represents serum sodium concentration at arginine vasopressin treatment start. Red box shows lowest serum sodium concentration during arginine vasopressin treatment. Solid lines are median, 25^th^ and 75^th^ quantile, and whiskers equal to 25^th^ quantile − 1.5 IQR and 75^th^ quantile + 1.5 IQR.

**Figure 2 children-11-01359-f002:**
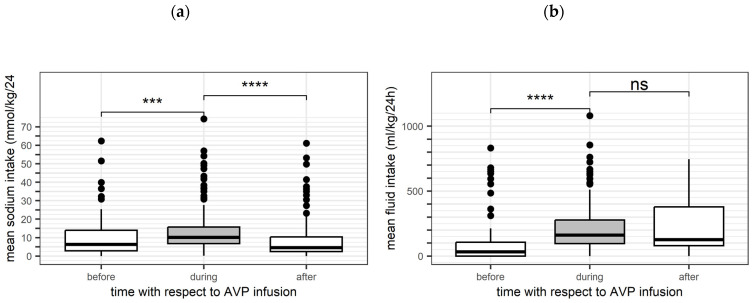
Amount of sodium and fluid uptake supply 24 h before, during, and 24 h after arginine vasopressin therapy: (**a**) sodium uptake over time presented as box plots (pairwise comparisons: ***: *p* = 0.00016, ****: *p* = 2.1 × 10^−9^); (**b**) fluid uptake over time presented as box plots (pairwise comparisons: ****: *p* < 2.22 × 10^−16^, ns: *p* = 0.36). Boxes show the interquartile range (IQR). Solid lines are the median, 25^th^ and 75^th^ quantile, and whiskers equal to 25^th^ quantile − 1.5 IQR and 75^th^ quantile + 1.5 IQR.

**Figure 3 children-11-01359-f003:**
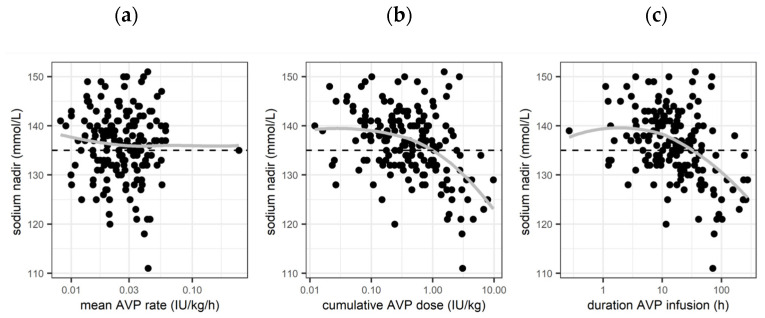
Association of lowest serum sodium concentration on various factors of arginine vasopressin therapy: (**a**) association of lowest serum sodium concentration with mean AVP rate; (**b**) association of lowest serum sodium concentration with cumulative AVP dose; (**c**) association of lowest serum sodium concentration with duration of AVP infusion. Grey lines: non-parametric smoothing line (loess). Horizontal black dashed line: sodium of 135 mmol/L (limit of hyponatremia).

**Figure 4 children-11-01359-f004:**
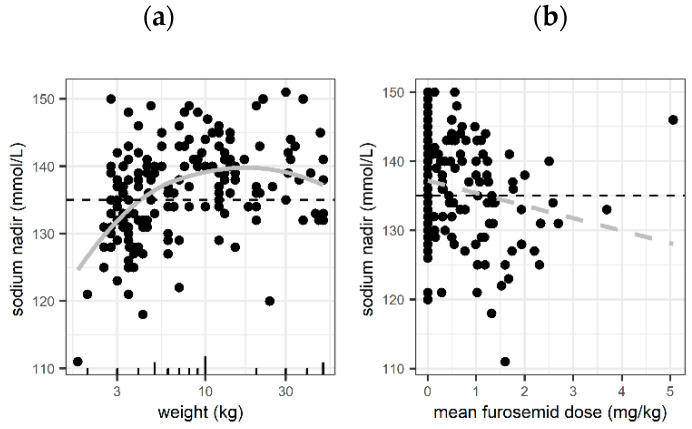
Lowest serum sodium concentration stratified by variables that remained significantly associated in multivariable analysis: (**a**) association of the lowest serum sodium concentration with patients’ weight (**b**) association of the lowest serum sodium concentration with mean furosemide dose. Grey solid line: non-parametric smoothing line (loess), grey dashed line: linear regression line. Horizontal black dashed line: sodium of 135 mmol/L (limit of hyponatremia).

**Table 1 children-11-01359-t001:** Baseline characteristics of the 170 patients included in the analysis.

Patient Characteristics	Value
Female Gender n (%)	75 (44.1)
Age (months)	4 [0, 33]
Weight (kg)	6.0 [3.5, 13.0]
Height (cm)	68 [53, 94]
Surgery n (%)	157 (92.4)
Invasive ventilation (h)	152 [40, 331]
LOS (d)	20.9 [8.8, 48.2]
Paediatric Index of Mortality II (%)	5.9 [2.7, 18.7]
Death n (%)	49 (28.8)
Main diagnosis	
Cardiac	118 (69.4%)
Shock	23 (15.5%)
Visceral	8 (4.7%)
Respiratory	5 (2.9%)
Trauma	3 (1.8%)
Other	13 (7.6%)
Vasopressin:	
Treatment duration (h)	13.6 h [6.2–32.6]
Cumulative dose (IU/kg)	0.37 [0.15–0.95]
Mean individual dose rate (IU/kg/h)	0.027 IU/kg/h [0.019–0.036]
Comedication during AVP administration:	
Adrenaline (µg/kg/min), n = 169 (99.4%)	0.07 [0.00, 0.12]
Noradrenaline (µg/kg/min), n = 170 (100%)	0.12 [0.05, 0.2]
Dopamine (µg/kg/min), n = 8 (4.7%)	0 [0.0, 4.8]
Dobutamine (µg/kg/min), n = 4 (2.4%)	0 [0.0, 0.45]
Hydrocortisone (mg/kg/d), n = 102 (60%)	3.8 [2.0, 6.0]
Dexamethasone (mg/kg/d), n = 3 (1.8%)	0.6 [0.6, 0.8]
Methylprednisolone (mg/kg/d), n = 15 (8.8%)	2.0 [1.0, 3.6]
Prednisolone (mg/kg/d), n = 4 (2.4%)	2.0 [2.0, 2.2]
Furosemide (mg/kg/d), n = 86 (50.6%)	1.0 [0.5, 1.6]
Spironolactone (mg/kg/d), n = 14 (8.2%)	1.5 [1.1, 1.8]
Hydrochlorothiazide (mg/kg/d), n = 4 (2.4%)	1.8 [1.4, 3.6]

Baseline characteristics are presented as median [interquartile range] for non-parametrically distributed continuous variables or as mean (standard deviation) for parametrically distributed variables. Categorial variables are presented as n (%).

## Data Availability

All data generated or analysed during this study are included in this article and in the Appendix A. Further enquiries can be directed to the corresponding author.

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
