# Peer review of "Serum Sodium Concentration During Arginine Vasopressin Infusion in Critically Ill Children"

_children, 2024, doi:10.3390/children11111359_

Round 1

Reviewer 1 Report

Comments and Suggestions for Authors

Abstract

In this retrospective analysis performed at a 25-bed tertiary paediatric intensive care unit patients were included if they were treated with intravenous arginine vasopressin between 2016 and 2022. Please rewrite this this sentence with reference to patients, not beds!

To clarify the mean arginine vasopressin dose rate was 0.027 U/kg/h [0.019-0.036]???

Doses are usually calculated in IU/min (international units)

The administration and monitoring of electrolytes and sodium is a routine practice in critical patients in intensive care. The importance of monitoring and the necessity of sodium administration in pediatric patients must be emphasized, since the doses are not commo

Introduction

There are 3 known vasopressin receptors V1, V2, V3, please rewrite(García-Álvarez, R.; Arboleda-Salazar, R. Vasopressin in Sepsis and Other Shock States: State of the Art. J. Pers. Med. 202313, 1548. https://doi.org/10.3390/jpm13111548)    line 57-65

Lines 93-97 are not very clear, please rewrite them. Please rewrite the inclusion and exclusion criteria clearly  (line 91-97)

Of course, since the inclusion criteria are not very clear, the results are unclear.   Please rewrite line 152-156

Please explain to the discussions if the administration

Hydrocortisone, Dexamethasone, Methylprednisolone, Prednisolone, Furosemide, Spironolactone, Hydrochlorothiazide, could they influence the plasma level of sodium according to the data obtained??

The limits of the study and the conclusions are good

The bibliography complies with the editing requirements, but can be easily increased

Author Response

Abstract

  1. In this retrospective analysis performed at a 25-bed tertiary paediatric intensive care unit patients were included if they were treated with intravenous arginine vasopressin between 2016 and 2022. Please rewrite this this sentence with reference to patients, not beds!

Authors` response: We are grateful for this input, the first sentence was change accordingly. It now reads: “In this retrospective analysis performed at a tertiary care paediatric intensive care unit with 2000 annual admissions patients were included if they were treated with intravenous vasopres-sin between 2016 and 2022.” Likewise, we added the annual admission number to “2. Methods 2.1. Settings and Participants” in line 92 and deleted bed-count. We deleted the sentence “Median length of intensive care stay […]” to reduce text and to stay within the word count limit. All line numbers provided in this rebuttal correspond to the track change version of the manuscript.

  1. To clarify the mean arginine vasopressin dose rate was 0.027 U/kg/h [0.019-0.036]??? Doses are usually calculated in IU/min (international units).

Authors` response: Thank you for this input. Paediatric Vasopressin prescriptions are weight based, they can be ordered in IU/kg/h or IU/kg/min [1]. As our electronic patient data management system works with IU/kg/h, we would like to keep this weight based unit. However, we changed wording to IU/kg or IU/kg/h throughout the document and updated Fig 3 on this.

  1. The administration and monitoring of electrolytes and sodium is a routine practice in critical patients in intensive care. The importance of monitoring and the necessity of sodium administration in pediatric patients must be emphasized, since the doses are not common

Authors` response: As this refers to the abstract with limited text space, we carefully modified the last sentence of the abstract conclusion to account for the reviewer`s input. It now reads: “This emphasises the need of close electrolyte monitoring and sodium substitution in children and adolescents under arginine vasopressin treatment to avoid hyponatraemia and related sequelae.”

Introduction

  1. There are 3 known vasopressin receptors V1, V2, V3, please rewrite(García-Álvarez, R.; Arboleda-Salazar, R. Vasopressin in Sepsis and Other Shock States: State of the Art. J. Pers. Med. 2023, 13, 1548. https://doi.org/10.3390/jpm13111548) line 57-65

Authors` response: We are grateful for adding this to the list of physiologic actions. We inserted the following sentence into lines 60/61: “Further, V3 AVP receptors in the anterior pituitary and in the pancreas steer the secretion of adrenocorticotropic hormone and insulin, respectively” and included the provided reference.

  1. Lines 93-97 are not very clear, please rewrite them. Please rewrite the inclusion and exclusion criteria clearly (line 91-97).

Authors` response: Indeed, this is an important input. Lines 95-107 were rewritten and in- and exclusion criteria explicitly stated.

  1. Of course, since the inclusion criteria are not very clear, the results are unclear. Please rewrite line 152-156

Authors` response: The reviewer was right, the lines 172-182 were restructured and rewritten to be clear on included and excluded patients.

  1. Please explain to the discussions if the administration Hydrocortisone, Dexamethasone, Methylprednisolone, Prednisolone, Furosemide, Spironolactone, Hydrochlorothiazide, could they influence the plasma level of sodium according to the data obtained??

Authors` response: In paragraph “3.3.3. Association of other clinical parameters with lowest serum sodium concentration” of the originally submitted manuscript we presented the associations of clinical parameters with lowest sodium levels as calculated in a multivariable regression analysis. Apart from AVP treatment duration we found patient and furosemide dose to be associated with lowest serum sodium concentrations. Steroid administration and doses showed no significant association in the multivariable regression analysis. In the results section we originally mentioned this in lines 249-255, in Figure 4, and discussed this briefly in “4. Discussion, 4.1 Intravenous AVP therapy and hyponatraemia” in lines 312-314. We acknowledge, that we did not present the complete regression analyses results and that this could be enhanced. Therefore, we added the full results into lines 293-296 and, further, added a short paragraph on mechanisms of action and discussion of possible influences of steroids and diuretics into the discussion in lines 352-363.

  1. The limits of the study and the conclusions are good

Authors` response: We thank the reviewer for this view on limitations and conclusions.

  1. The bibliography complies with the editing requirements, but can be easily increased.

Authors` response: We thank the reviewer also for this statement and would like to point out, that we updated the bibliography according to the suggestions of another reviewer.

References

  1. Choong, K. Vasopressin in Pediatric Critical Care. Journal of pediatric intensive care 2016, 5, 182-188, doi:10.1055/s-0036-1583282.

Reviewer 2 Report

Comments and Suggestions for Authors

Regarding the article entitled "Serum Sodium Concentration during Arginine Vasopressin Infusion in Critically Ill Children" I would like to congratulate the authors for increasing awareness of one of the potential adverse effects of the increasingly widespread use of arginine vasopressin. This adverse effect is often underestimated due to the critical situation of the patients receiving this vasopressor.

Abstract: Adequate.

Introduction:

From my point of view, the introduction on the severity of shock states and their consequences (lines 41 - 45) is omittable. This manuscript does not discuss the need to treat hypotension, which can have serious consequences in our patients, but rather the potential consequences of the need to begin treatment with vasopressin.

The authors point out that "organ perfusion and thus oxygen and nutrient transport to the organs can be restored [3-7]" (lines 47 - 48). In my opinion, this sentence does not need so many bibliographical references.

About the following sentence: "It is increasingly used in the routine treatment of critically ill children and adults [5,11-19]" (lines 54-55), from my point of view, authors should focus the introduction more clearly on the target population: critically ill children

The second paragraph of the introduction discusses the pathophysiology behind the use of arginine vasopressin. I consider that it can be summarized to make it easier for potential readers to read.

Methods:

Authors must follow the checklist corresponding to retrospective observational studies. Therefore, the manuscript should be modified in accordance with this checklist. In case the authors have followed this checklist, it should be explicitly indicated and included in the review of the manuscript.

There are data that should be included in the analysis that may affect the serum sodium concentration, such as parenteral nutrition or the type and amount of fluid therapy administered during admission to the unit or in the preoperative period. I also recommend that they be more specific in the diagnosis of admission to the unit, since there are concomitant pathologies (diarrhea, heart failure, vomiting, renal disorders, etc.) that can alter the natremia of the patients.

Results:

The authors state that "In total 181 patients with AVP administration and a median age of 5 months [0-34] signed general consent" (lines 154-155). I don't know if I understand this sentence correctly: The consents must have been signed by their parents or legal guardians? As this is a retrospective study, how could they sign the informed consent? I think this (methodological/ethical) data should be better explained or clarified.

Regarding the course of sodium in these patients, I have a question: how can the authors rule out other causes of the decrease in sodium in these patients? Are we sure that the decrease in sodium in these patients is due only to the use of arginine vasopressin?

In table 1 they present the baseline characteristics of the patients, among which is "Death". I consider that mortality is an outcome rather than a baseline characteristic.

Of the 170 patients included in table 1, we only have the baseline values ​​of 95 patients "(available for a subset of 95 patients)" (lines 168 - 169). How can the authors know if the sodium of the 170 patients decreased during the therapy if they lack the data of almost half of the sample?

On the other hand, the sodium concentrations at 20-28 hours after treatment with AVP are taken only from a sample of 116 patients "(available for n=116 patients)" (line 172). In my opinion, the analysis performed should be reviewed, as they found statistically significance (p = 0.012), but I am not sure that they come from the same population. I recommend restructuring the included population (lines 168 - 179). Perhaps the authors only have the complete data from 95 patients and the results ​​changed and were no longer statistically significant.

I recommend performing a more specific multivariate analysis to be able to be more specific between the relationship of treatment and the evolution of plasma sodium concentrations.

Discussion:

I recommend avoiding repeating information already provided in the introduction. In my view, the information provided in lines 257 - 259 and in lines 260 - 266 is superimposable to that provided in the introduction of the manuscript.

References:

I think references should be reviewed, since 24 of the 31 references are at least 10 years old.

Author Response

Regarding the article entitled "Serum Sodium Concentration during Arginine Vasopressin Infusion in Critically Ill Children" I would like to congratulate the authors for increasing awareness of one of the potential adverse effects of the increasingly widespread use of arginine vasopressin. This adverse effect is often underestimated due to the critical situation of the patients receiving this vasopressor.

Abstract

  1. Adequate.

Authors` response: We thank the reviewer for this appraisal.

Introduction

  1. From my point of view, the introduction on the severity of shock states and their consequences (lines 41 - 45) is omittable. This manuscript does not discuss the need to treat hypotension, which can have serious consequences in our patients, but rather the potential consequences of the need to begin treatment with vasopressin.

Authors` response: We agree with the reviewer and modified the introductory passages in the requested way.

  1. The authors point out that "organ perfusion and thus oxygen and nutrient transport to the organs can be restored [3-7]" (lines 47 - 48). In my opinion, this sentence does not need so many bibliographical references.

Authors` response: We agree and reduced the citation count to one.

  1. About the following sentence: "It is increasingly used in the routine treatment of critically ill children and adults [5,11-19]" (lines 54-55), from my point of view, authors should focus the introduction more clearly on the target population: critically ill children

Authors` response: We deleted statements on adults and deleted related references accordingly.

  1. The second paragraph of the introduction discusses the pathophysiology behind the use of arginine vasopressin. I consider that it can be summarized to make it easier for potential readers to read.

Authors` response: We added the V3-receptor mediated mechanism on request of another reviewer and deleted superfluous information in lines 60-68. All line numbers provided in this rebuttal correspond to the track change version of the manuscript

Methods

  1. Authors must follow the checklist corresponding to retrospective observational studies. Therefore, the manuscript should be modified in accordance with this checklist. In case the authors have followed this checklist, it should be explicitly indicated and included in the review of the manuscript.

Authors` response: We followed the STROBE checklist for reporting retrospective data (https://www.strobe-statement.org/checklists/) and inserted this as statement into lines 168-169.

  1. There are data that should be included in the analysis that may affect the serum sodium concentration, such as parenteral nutrition or the type and amount of fluid therapy administered during admission to the unit or in the preoperative period.

Authors` response: This is an important point. In the original analysis we included all supplemental sodium that was administered in addition to baseline nutrition. We assumed, that baseline nutrition without supplements should not lead to hypo- or hypernatraemia. Indeed, this was not exactly specified in the initial manuscript. We added a statement into lines 130-136. Further, we explained, that we counted the fluids as total fluid volume uptake per day. We refrained from analysing every single liquid type, because we felt, that the exact formula is less important than their sodium content actually taken up.

  1. I also recommend that they be more specific in the diagnosis of admission to the unit, since there are concomitant pathologies (diarrhea, heart failure, vomiting, renal disorders, etc.) that can alter the natremia of the patients.

Authors` response: Indeed, many patients had more than one diagnosis, some have many diagnoses. We did not want to analyse every single diagnosis, because we felt, that we would have too many diagnoses with very small numbers, if diagnoses would have been grouped in a more diverse fashion and it would not be possible to weigh the relevance of different diagnoses in each patients` diagnosis list. Instead, we took the main (and therefore most important) diagnosis, which lead to PICU admission, a procedure often used for grouping patients into subgroups. We judged this to be the best indicator for the patients` main pathology and therefore for the most important health problem. Initially this was mentioned as “primary diagnosis leading to ICU admission” in lines 116-117 of the original version and would refrain from a change here. For full disclosure, we provide the full list of all primary diagnosis in this document, but these are indeed many different diagnoses, which does not shed any new light on patients in relation to their sodium course. We are open to include them as supplement, if requested.

Detailed list of primary diagnosis leading to ICU admission for this reviewer`s information:

UID

Diagnosis

Category

4

Truncus arteriosus communis Type 1

Cardiac

5

Terminal heart failure in non-compaction cardiomyopathy

Cardiac

6

Cardiogenic shock in myocarditis (need for ECMO)

Cardiac

9

Bilateral AV valve insufficiency

Cardiac

11

Cardiopulmonary resuscitation with double outlet right ventricle

Cardiac

12

Pulmonary hypertensive crises

Cardiac

13

Severe dilated cardiomyopathy

Cardiac

18

Cardiac arrest

Cardiac

19

Heart failure with tricuspid regurgitation

Cardiac

20

Univentricular heart

Cardiac

21

Hypoplastic left heart syndrome

Cardiac

22

Multiple heart defects (including AV and VA discordance)

Cardiac

24

Double inlet left ventricle

Cardiac

25

Pulmonary atresia with intact ventricular septum

Cardiac

27

Pulmonary atresia with ventricular septal defect

Cardiac

28

Severe subaortic stenosis

Cardiac

30

Congenitally corrected transposition of the great arteries

Cardiac

31

Borderline Left Ventricle

Cardiac

32

Single ventricle with double outlet right ventricle

Cardiac

33

Pulmonary atresia with intact ventricular septum

Cardiac

34

Severe heart failure in hypoplastic left heart syndrome

Cardiac

37

Balanced AVSD and severe pulmonary hypertension

Cardiac

38

Univentricular heart with tricuspid valve atresia

Cardiac

39

Hypoplastic left heart syndrome

Cardiac

40

Tricuspid atresia with small ventricular septal defect

Cardiac

41

Cardiac arrest, aspiration event

Cardiac

42

Borderline left ventricle

Cardiac

46

Dysbalanced atrioventricular septal defect

Cardiac

47

Pulmonary atresia with ventricular septal defect

Cardiac

48

Tetralogy of Fallot

Cardiac

49

d-transposition of the great arteries

Cardiac

51

Pulmonary atresia with ventricular septal defect

Cardiac

52

Tetralogy of Fallot

Cardiac

56

Unbalanced atrioventricular canal with hypoplastic left ventricle

Cardiac

57

Hypoplastic left heart syndrome

Cardiac

58

Cardiogenic shock of unknown origin

Cardiac

59

Serial right ventricular stenosis

Cardiac

61

Hypoplastic left heart syndrome

Cardiac

62

d-transposition of the large arteries

Cardiac

65

Pulmonary atresia with ventricular septal defect

Cardiac

67

Total pulmonary vein malformation

Cardiac

68

High-grade stenosis of pulmonary artery banding

Cardiac

73

Hypoplastic left heart syndrome

Cardiac

74

Double outlet right ventricle

Cardiac

76

Perinatal cardiac arrest

Cardiac

77

Shone-complex with borderline left ventricle

Cardiac

78

Hypoplastic right heart

Cardiac

79

Double outlet right ventricle

Cardiac

80

Hypoplastic left heart syndrome

Cardiac

81

Hypoplastic left heart syndrome

Cardiac

82

Left isomerism, mesocardia

Cardiac

83

Tricuspid atresia

Cardiac

85

Complete balanced atrioventricular canal

Cardiac

86

Left ventricular outflow tract obstruction

Cardiac

87

Membranous pulmonary atresia

Cardiac

88

Total pulmonary vein malformation of the supracardiac type with obstruction

Cardiac

89

Progressive valvular and supravalvular pulmonary stenosis

Cardiac

90

Borderline left ventricle

Cardiac

91

Severe mitral valve insufficiency

Cardiac

92

Hypoplastic left heart syndrome

Cardiac

95

Pulmonary atresia with intact ventricular septum

Cardiac

96

Hypoplastic left heart syndrome and severe tricuspid regurgitation

Cardiac

98

Cardiogenic shock

Cardiac

100

Increasing dysfunction of the ventricular electrode

Cardiac

101

Severe residual left-sided atrioventricular valve stenosis

Cardiac

103

Persistent pulmonary hypertension with meconium aspiration

Cardiac

104

Double outlet right ventricle

Cardiac

105

d-transposition of the great arteries

Cardiac

107

Truncus arteriosus communis

Cardiac

108

Fallot tetralogy

Cardiac

110

Double outlet right ventricle Fallot type

Cardiac

112

Pulmonary atresia with ventricular septal defect

Cardiac

113

Acute decompensated heart failure

Cardiac

115

Shone complex

Cardiac

116

Interrupted aortic arch

Cardiac

118

Interrupted aortic arch type B with lusorian artery

Cardiac

119

Membranous pulmonary atresia with ventricular septal defect

Cardiac

120

Double outlet of the right ventricle and transposition of the great arteries

Cardiac

121

d-transposition of the great arteries, residual ventricular septal defect

Cardiac

122

Tricuspid atresia type 1b and syndromic underlying disease

Cardiac

123

Hemi truncus arteriosus and pulmonary hypertension

Cardiac

124

Transposition of the large arteries

Cardiac

125

Fallot tetralogy

Cardiac

127

Postpartum asphyxia

Cardiac

129

Critical valvular aortic stenosis in Norwood 1 circulation

Cardiac

130

Pentalogy of Fallot

Cardiac

132

d-transposition of the great arteries

Cardiac

133

Corrected large perimembranous malalignment ventricular septal defect

Cardiac

135

Left ventricular hypoplasia and truncus arteriosus communis type 2b

Cardiac

136

Double outlet right ventricle and Kabuki syndrome

Cardiac

137

Cardiogenic shock

Cardiac

138

d-transposition of the great arteries, ventricular septal defect

Cardiac

139

Hypoplastic right heart syndrome and thrombotic occlusion

Cardiac

140

Membranous pulmonary atresia with ventricular septal defect

Cardiac

141

Valvular and subvalvular aortic stenosis

Cardiac

142

Persistent patent ductus arteriosus

Cardiac

143

Atrial septal defect

Cardiac

144

Shone-like complex

Cardiac

145

Unbalanced atrioventricular septal defect

Cardiac

147

Total correction of pulmonary atresia

Cardiac

148

Malposition of the large arteries

Cardiac

149

Double inlet left ventricle

Cardiac

151

Atypical pulmonary vein mouth and pulmonary vein stenosis

Cardiac

152

Cardiopulmonary resuscitation, corrected tetralogy of fallot

Cardiac

153

Medium-sized perimembranous ventricular septal defect

Cardiac

154

Foramen ovale and persistent ductus arteriosus

Cardiac

155

d-transposition of the great arteries with intact ventricular septal defect

Cardiac

157

Truncus arteriosus communis type 2 and ventricular septal defect

Cardiac

158

Complex aortic coarctation and two ventricular septal defects

Cardiac

160

Noonan syndrome, severe valvular and supravalvular pulmonary stenosis

Cardiac

161

d-transposition of the great arteries, SIRS

Cardiac

162

Pulmonary atresia with large ventricular septal defect

Cardiac

163

Aortic isthmus stenosis with hypoplastic aortic arch

Cardiac

164

Shone Complex

Cardiac

167

Shone complex, post-interventional tricuspid regurgitation

Cardiac

168

Tetralogy of Fallot

Cardiac

172

Pulmonary atresia with multiple ventricular septal defects

Cardiac

174

Pulmonary atresia with ventricular septal defect

Cardiac

175

Single ventricle with double outlet Right ventricle

Cardiac

178

Double inlet left ventricle

Cardiac

179

Severe mitral stenosis and complete atrioventricular septal defect

Cardiac

180

Persistent foramen oval and ductus arteriosus

Cardiac

1

Decompensated septic shock

Shock

7

Decompensated shock with multiple organ failure

Shock

10

Septic shock after heart transplantation

Shock

14

Septic shock in necrotising fasciitis

Shock

16

Decompensated septic shock

Shock

29

Catecholamine-refractory septic shock

Shock

50

Decompensated shock

Shock

54

Toxic shock syndrome

Shock

55

Recompensated septic shock due to Pseudomonas aeruginosa

Shock

64

Decompensated septic shock due to Pseudomonas aeruginosa

Shock

69

Acute renal failure requiring dialysis

Shock

71

Gram negative sepsis

Shock

94

Toxic shock syndrome

Shock

97

Multi-organ failure with shock

Shock

117

Decompensated septic shock

Shock

146

Paediatric multisystem inflammatory syndrome

Shock

150

Decompensated septic shock

Shock

165

Haemolytic uraemic syndrome

Shock

176

Postoperative severe septic shock

Shock

177

Hypoxaemia in stenosis of a modified BT stent

Shock

26

Covered perforation in the area of the left colonic flexure

Visceral

63

Intestinal pseudo-obstruction

Visceral

72

Ileus

Visceral

102

Diaphragmatic hernia

Visceral

114

Congenital diaphragmatic hernia, necrotizing enterocolitis

Visceral

128

Midgut volvulus with total necrosis of the small intestine

Visceral

166

Loop volvulus with small bowel segment necrosis

Visceral

169

Congenital diaphragmatic hernia

Visceral

170

Ischaemic colitis in the descending colon

Visceral

36

Severe acute respiratory distress syndrome

Respiratory

111

Acute respiratory insufficiency

Respiratory

134

Surfactant deficiency and suspected neonatal infection

Respiratory

181

Acute respiratory distress syndrome RSV

Respiratory

2

Severe craniocerebral trauma

Trauma

35

Severe polytrauma due to road traffic accident

Trauma

60

Multi-organ failure in a high voltage accident

Trauma

109

Severe polytrauma after fall from 20 meters

Trauma

3

Severe multiple disabilities with chromosome 1, 4, 10 deletion

Other

15

Acute hypoxic ischaemic encephalopathy

Other

17

Hypoxic ischaemic encephalopathy

Other

44

Right isomerism with ambiguous thoracic and abdominal sinus

Other

45

Heterotaxia syndrome, postoperative arrhythmia

Other

66

Recurrence of the rupture of an aneurysm of the pericallosal artery

Other

70

Jacobsen syndrome

Other

99

Acute intracerebral haemorrhage of a cerebellar aneurysm

Other

106

Fetal repair of a myelomeningocele

Other

156

AV malformation in the cingulate gyrus with intracerebral haemorrhage

Other

159

Thrombotic occlusion of the proximal external iliac artery

Other

171

Partial trisomy chromosome 6 and neonatal hypoglycaemia

Other

Results

  1. The authors state that "In total 181 patients with AVP administration and a median age of 5 months [0-34] signed general consent" (lines 154-155). I don't know if I understand this sentence correctly: The consents must have been signed by their parents or legal guardians? As this is a retrospective study, how could they sign the informed consent? I think this (methodological/ethical) data should be better explained or clarified.

Authors` response: The authors are grateful for this hint, this was not explicitly stated in the first version. All patients and/or their legal guardians, who are hospitalized in the University Children`s Hospital Zurich, are always asked for their written consent for further (coded) use of the routine data stored in the electronic health care records of the hospital. This was true for 181 of 244 eligible patients. As the reviewer is right, this was a retrospective study and we did not ask for consent again. Nevertheless, we asked the local legal authorities for approval of this approach and approval was granted by the cantonal ethics committee on 10th of December 2021 under the license number KEK 2021-02276. An explanatory passage about general informed consent was inserted in lines 99-102.

  1. Regarding the course of sodium in these patients, I have a question: how can the authors rule out other causes of the decrease in sodium in these patients? Are we sure that the decrease in sodium in these patients is due only to the use of arginine vasopressin?

Authors` response: Of course, we cannot be 100% sure, that we missed important influence factors. However, during the planning phase of this study we reflected on every possible influence factor and included the following variables into the univariable and multivariable regression analyses: age weight, primary diagnosis leading to ICU admission, PIM II, AVP dose and treatment duration, sodium supplementation in addition to nutrition, daily intravenous fluids, steroid, and diuretic doses. These results were omitted so far and we included them into the text into lines 293-296.

  1. In table 1 they present the baseline characteristics of the patients, among which is "Death". I consider that mortality is an outcome rather than a baseline characteristic.

Authors` response: In our eyes, death can serve as baseline characteristic, outcome parameter, or study endpoint, depending on the specific study objectives. If researchers retrospectively examine a cohort without defining death as outcome parameter (e.g. death events versus survival during the study period) or e.g. reduction of mortality through a therapeutic intervention as study endpoint, the number of deaths in the cohort can serve as a descriptive surrogate parameter for severity of disease. In our case, we retrospectively looked at a group of patients with a very high mortality rate (28.8%) for the PICU cohort in our hospital: we usually have around 50 deaths/year among a total of 2000 annual admissions (=2.5%). We did not use interventions to reduce mortality and we did not statistically compared the study cohort`s mortality against e.g. patients who were not treated with AVP. Death was only used to characterize the sickness of the study cohort and the death rate was not influenceable anymore at the time of analysis. We did not expect that we can claim a statement on the degree of influence of AVP on death from this analysis. Therefore, we would like to keep it as descriptive baseline parameter. We used this approach in a recent study on copeptin: https://pmc.ncbi.nlm.nih.gov/articles/PMC9222164/pdf/children-09-00794.pdf.

  1. Of the 170 patients included in table 1, we only have the baseline values ​​of 95 patients "(available for a subset of 95 patients)" (lines 168 - 169). How can the authors know if the sodium of the 170 patients decreased during the therapy if they lack the data of almost half of the sample? On the other hand, the sodium concentrations at 20-28 hours after treatment with AVP are taken only from a sample of 116 patients "(available for n=116 patients)" (line 172). In my opinion, the analysis performed should be reviewed, as they found statistically significance (p = 0.012), but I am not sure that they come from the same population. I recommend restructuring the included population (lines 168 - 179). Perhaps the authors only have the complete data from 95 patients and the results ​​changed and were no longer statistically significant.

Authors` reply: Here we agree, that we do not have full data sets for the time frame of 24 h before start of AVP. This is a limitation of a retrospective study design. In many study cases, the patients were admitted post-surgery to the ICU and we just do not have all comparable data on sodium from 24 h before treatment start, because they were not taken. We did search for them, but we only found data for 95 patients. We acknowledge the reviewer`s concerns and performed a sensitivity analyses with 59 patients with complete data before, during, and after AVP treatment. We created a Supplemental Material file and added the results of the sensitivity analysis as Table. Further, we restructured the results section. We added a statement on missing data in methods (no data imputation, lines 153-154). We added a statement on missings as limitation into lines 394-397.

  1. I recommend performing a more specific multivariate analysis to be able to be more specific between the relationship of treatment and the evolution of plasma sodium concentrations.

Authors` reply: As mentioned above, we performed an exhaustive regression analysis and added the results into lines 293 – 296.

Discussion

  1. I recommend avoiding repeating information already provided in the introduction. In my view, the information provided in lines 257 - 259 and in lines 260 - 266 is superimposable to that provided in the introduction of the manuscript.

Authors` reply: We shortened the introductory phrases in section “4.1 Intravenous AVP therapy and hyponatraemia” and deleted information already provided in the Introduction. We agree, that every information should only be provided once and that redundancy should be avoided.

References

  1. I think references should be reviewed, since 24 of the 31 references are at least 10 years old.

Authors` reply: We reviewed the references, deleted redundancy, and added new references, where appropriate.

The following references were deleted:

  1. Tweddell, J.S.; Hoffman, G.M.; Fedderly, R.T.; Ghanayem, N.S.; Kampine, J.M.; Berger, S.; Mussatto, K.A.; Litwin, S.B. Patients at risk for low systemic oxygen delivery after the Norwood procedure. Ann Thorac Surg 2000, 69, 1893-1899, doi:10.1016/s0003-4975(00)01349-7.
  2. Wernovsky, G.; Kuijpers, M.; Van Rossem, M.C.; Marino, B.S.; Ravishankar, C.; Dominguez, T.; Godinez, R.I.; Dodds, K.M.; Ittenbach, R.F.; Nicolson, S.C., et al. Postoperative course in the cardiac intensive care unit following the first stage of Norwood reconstruction. Cardiol Young 2007, 17, 652-665, doi:10.1017/s1047951107001461.
  3. Killinger, J.S.; Hsu, D.T.; Schleien, C.L.; Mosca, R.S.; Hardart, G.E. Children undergoing heart transplant are at increased risk for postoperative vasodilatory shock. Pediatr Crit Care Med 2009, 10, 335-340, doi:10.1097/PCC.0b013e3181a316c0.
  4. Kissoon, N.; Orr, R.A.; Carcillo, J.A. Updated American College of Critical Care Medicine--pediatric advanced life support guidelines for management of pediatric and neonatal septic shock: relevance to the emergency care clinician. Emerg. Care 2010, 26, 867-869, doi:10.1097/PEC.0b013e3181fb0dc0.
  5. Podrid, P.J.; Fuchs, T.; Candinas, R. Role of the sympathetic nervous system in the genesis of ventricular arrhythmia. Circulation 1990, 82, I103-113.
  6. Cheung, P.Y.; Barrington, K.J.; Pearson, R.J.; Bigam, D.L.; Finer, N.N.; Van Aerde, J.E. Systemic, pulmonary and mesenteric perfusion and oxygenation effects of dopamine and epinephrine. Am J Respir Crit Care Med 1997, 155, 32-37, doi:10.1164/ajrccm.155.1.9001285.
  7. Dünser, M.W.; Mayr, A.J.; Ulmer, H.; Knotzer, H.; Sumann, G.; Pajk, W.; Friesenecker, B.; Hasibeder, W.R. Arginine vasopressin in advanced vasodilatory shock: a prospective, randomized, controlled study. Circulation 2003, 107, 2313-2319, doi:10.1161/01.Cir.0000066692.71008.Bb.
  8. Rosenzweig, E.B.; Starc, T.J.; Chen, J.M.; Cullinane, S.; Timchak, D.M.; Gersony, W.M.; Landry, D.W.; Galantowicz, M.E. Intravenous arginine-vasopressin in children with vasodilatory shock after cardiac surgery. Circulation 1999, 100, Ii182-186, doi:10.1161/01.cir.100.suppl_2.ii-182.
  9. Lechner, E.; Hofer, A.; Mair, R.; Moosbauer, W.; Sames-Dolzer, E.; Tulzer, G. Arginine-vasopressin in neonates with vasodilatory shock after cardiopulmonary bypass. Eur J Pediatr 2007, 166, 1221-1227, doi:10.1007/s00431-006-0400-0.
  10. Jerath, N.; Frndova, H.; McCrindle, B.W.; Gurofsky, R.; Humpl, T. Clinical impact of vasopressin infusion on hemodynamics, liver and renal function in pediatric patients. Intensive Care Med 2008, 34, 1274-1280, doi:10.1007/s00134-008-1055-2.
  11. Mastropietro, C.W.; Clark, J.A.; Delius, R.E.; Walters, H.L., 3rd; Sarnaik, A.P. Arginine vasopressin to manage hypoxemic infants after stage I palliation of single ventricle lesions. Pediatr Crit Care Med 2008, 9, 506-510, doi:10.1097/PCC.0b013e3181849ce0.
  12. Morrison, W.E.; Simone, S.; Conway, D.; Tumulty, J.; Johnson, C.; Cardarelli, M. Levels of vasopressin in children undergoing cardiopulmonary bypass. Cardiol Young 2008, 18, 135-140, doi:10.1017/s1047951108001881.
  13. Burton, G.L.; Kaufman, J.; Goot, B.H.; da Cruz, E.M. The use of Arginine Vasopressin in neonates following the Norwood procedure. Cardiol Young 2011, 21, 536-544, doi:10.1017/s1047951111000370.
  14. Serpa Neto, A.; Nassar, A.P.; Cardoso, S.O.; Manetta, J.A.; Pereira, V.G.; Espósito, D.C.; Damasceno, M.C.; Russell, J.A. Vasopressin and terlipressin in adult vasodilatory shock: a systematic review and meta-analysis of nine randomized controlled trials. Crit Care 2012, 16, R154, doi:10.1186/cc11469.
  15. Thibonnier, M. Signal transduction of V1-vascular vasopressin receptors. Regul Pept 1992, 38, 1-11, doi:10.1016/0167-0115(92)90067-5.
  16. Zingg, H.H. Vasopressin and oxytocin receptors. Baillieres Clin Endocrinol Metab 1996, 10, 75-96, doi:10.1016/s0950-351x(96)80314-4.

The following references were added:

  1. De Backer, D.; Foulon, P. Minimizing catecholamines and optimizing perfusion. Crit Care 2019, 23, 149, doi:10.1186/s13054-019-2433-6.
  2. Bankir, L.; Bichet, D.G.; Morgenthaler, N.G. Vasopressin: physiology, assessment and osmosensation. J Intern Med 2017, 282, 284-297, doi:10.1111/joim.12645.
  3. Escudero, V.J.; Mercadal, J.; Molina-Andújar, A.; Piñeiro, G.J.; Cucchiari, D.; Jacas, A.; Carramiñana, A.; Poch, E. New Insights Into Diuretic Use to Treat Congestion in the ICU: Beyond Furosemide. Front Nephrol 2022, 2, 879766, doi:10.3389/fneph.2022.879766.
  4. Pirracchio, R.; Annane, D.; Waschka, A.K.; Lamontagne, F.; Arabi, Y.M.; Bollaert, P.E.; Billot, L.; Du, B.; Briegel, J.; Cohen, J.; et al. Patient-Level Meta-Analysis of Low-Dose Hydrocortisone in Adults with Septic Shock. NEJM Evid 2023, 2, EVIDoa2300034, doi:10.1056/EVIDoa2300034.

Authors` general comment on reviewer #2: In general we would like to thank this reviewer for the thoroughness of this review. We saw that reviewer #2 invested a considerable amount of time to provide useful and expedient input to improve the article. We are convinced, that we could address most of the points criticized and are open for further discussions.

Round 2

Reviewer 2 Report

Comments and Suggestions for Authors

The authors have responded to all issues identified in the previous review.